# Contributions of Distribution Modelling to the Ecological Study of Psittaciformes

**José R. Ferrer-Paris** [1,2] and **Ada Sánchez-Mercado** [1,3,*]

1   School of Biological, Earth and Environmental Sciences, University of New South Wales, Kensington, NSW 2052, Australia; j.ferrer@unsw.edu.au
2   Data Science Hub, University of New South Wales, Kensington, NSW 2052, Australia
3   Escuela de Ciencias Ambientales, Universidad Espiritu Santo, Samborondon 092301, Ecuador
*   Correspondence: a.sanchez@unsw.edu.au; Tel.: +61-48-120-3171

**Abstract:** We provide an overview of the use of species distribution modeling to address research questions related to parrot ecology and conservation at a global scale. We conducted a literature search and applied filters to select the 82 most relevant studies to discuss. The study of parrot species distribution has increased steadily in the past 30 years, with methods and computing development maturing and facilitating their application for a wide range of research and applied questions. Conservation topics was the most popular topic (37%), followed by ecology (34%) and invasion ecology (20%). The role of abiotic factors explaining parrot distribution is the most frequent ecological application. The high prevalence of studies supporting on-ground conservation problems is a remarkable example of reduction in the research–action gap. Prediction of invasion risk and assessment of invasion effect were more prevalent than examples evaluating the environmental or economic impact of these invasions. The integration of species distribution models with other tools in the decision-making process and other data (e.g., landscape metrics, genetic, behavior) could even further expand the range of applications and provide a more nuanced understanding of how parrot species are responding to their even more changing landscape and threats.

**Keywords:** distribution; conservation; ecology; environmental niche modelling; research selection function; parrots; psittacids; species distribution models; state observation models

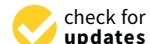



## 1. Parrots and Their Important Ecological Role

The order Psittaciformes (including parakeets, macaws, cockatoos, and allies, hereafter parrots) is a diverse order of birds with a wide range of morphological variations and foraging behaviors (~420 spp) [1]. Parrots can reach high density and biomass in many tropical and subtropical regions across the Americas, Africa, Europe, Asia and Oceanian regions [2]. The study of parrots' distribution patterns and factors driving them allows us to improve our understanding of their ecological role. With a wide diversity of biotic requirements (from generalist to specialist apex frugivores) and high prevalence within the bird community, parrots may have a broad influence on the structure of animal and plant communities and ecosystem functions [2–4].

Monitoring the distribution of parrot populations is also an important task for effective management and conservation of both threatened and problematic species [5]. Parrots are among the most threatened avian orders, with 46% of their species under threat and 56% of their populations experiencing population declines [6]. Abundance and distribution declines are driven by modification of their natural habitat and environment, in addition to nest poaching for the illegal market [7,8]. Human-modified environments are quickly encroaching on the most important areas for parrots in the Americas and Oceanian regions [4]; in Australia alone, parrots have lost at least 38% of their potential natural habitat [9].

Reduction in native parrot distribution is, however, only one side of the conservation problem. Parrots are among the most common companion animals, and intentional and

unintentional birds released from captivity have been related to the establishment of invasive parrot populations beyond their native distributional limits, causing damage to agriculture and natural environments [9,10].

Species distributions are complex biological phenomena, and many factors interact to determine a species' geographical range [11]. Due to the lack of extensive spatial records of occurrence, it is usually necessary to apply statistical methods to describe and predict species distribution. The key assumption of spatial analysis of wildlife populations is that spatial and temporal patterns in population state variables (i.e., occurrence, abundance or density, richness) represent the response of the species to underlying heterogeneity in external factors such as environment conditions and resource availability [12]. Interpretation of these patterns is scale dependent: at large spatial scales they reflect the overall constraints and conditions influencing species distribution, at intermediate scales they are related to population responses (including meta- and sub-populations), and at small spatial scales they can reflect individual behavioral responses [12].

Species distribution models have become an essential part of the analytical toolkit for ecological studies of birds. This is mainly due to the increasing accumulation and aggregation of basic biodiversity data (species location records), availability of worldwide abiotic environmental variables, and development of Geographic Information Systems [13]. Species distribution models use different algorithms and methods (e.g., MaxEnt, regressions, and occupancy models) to link field observations with spatially explicit explanatory or predictive variables. These variables can then be used to make spatial predictions that can be scaled up to whole landscapes or geographical regions.

Here, we provide an overview of the use of species distribution modeling to understand parrot distribution and place them in the broad conceptual context of the ecological scale at which spatial and temporal patterns are evaluated. For this, we combined quantitative methods of selection and analysis of scientific literature and a narrative discussion of the more relevant studies found. In Section 2, we use a structured search protocol to select relevant scientific literature and classify this sample of publications into a set of topics and general and specific applications. We quantify trends in publication rates and taxonomic and geographical coverage of these topics. In Section 3, we discuss how these methods have been applied to address research questions related to parrot ecology, conservation and biogeography, and in Section 4 we appraise to the extent to which emerging analytical tools have been implemented or could be exploited to explore new research questions in the future.

## 2. Literature Review of Distribution Modelling in Parrot Species

### 2.1. Sample of Scientific Literature

Our main objective was to provide a broad overview of the different topics of research. We limited our search to one search engine (Web of Science, WoS) and one language (English), and used a workflow to apply automatic and manual filters to detect the most relevant publications within the extracted list of references. Thus, although quantitative analyses were limited to a single sample of the literature, we used them to illustrate general trends and acknowledge their inherent biases and limitations. Although we do not explicitly compare this sample with other sources, we are confident that this search is representative of the overall trends in the scientific literature. Recent reviews have shown that among several academic search engines, WoS is more selective than Dimensions and GoogleScholar, and has a high degree of overlap with Scopus [14,15]. We are aware that the contribution of publications in non-English languages is high and by focusing our search on only English published papers we obtained a biased sample [16]. However, because several non-English journals include abstracts in English [17] we are confident that we were able to obtain a good representation of topics and applications published on parrot distribution research.

We conducted a literature search on the database Web of Science (WoS) using terms in English related with the focal taxonomic group (Psittaci*, parrot*, macaw*, parakeet*, amazon*, cockato*) in the themes section. This resulted in a total sample of 12,699 documents for the period 1900 to March 2020. At least 88% of the documents were originally published in English, 7.1% in 21 other languages, and 5% did not have information on the original language. Although our sample contains more than 12,000 scientific articles published in the last 100 years, there is a clear difference in the rate of documents per year for the periods before and after 1990 (Figure 1a). This may be an artifact of uneven coverage of the global literature in the WoS database; for example, a lack of digitalization of pre-1990 documents, or an increase in the number of sources included after 1990.

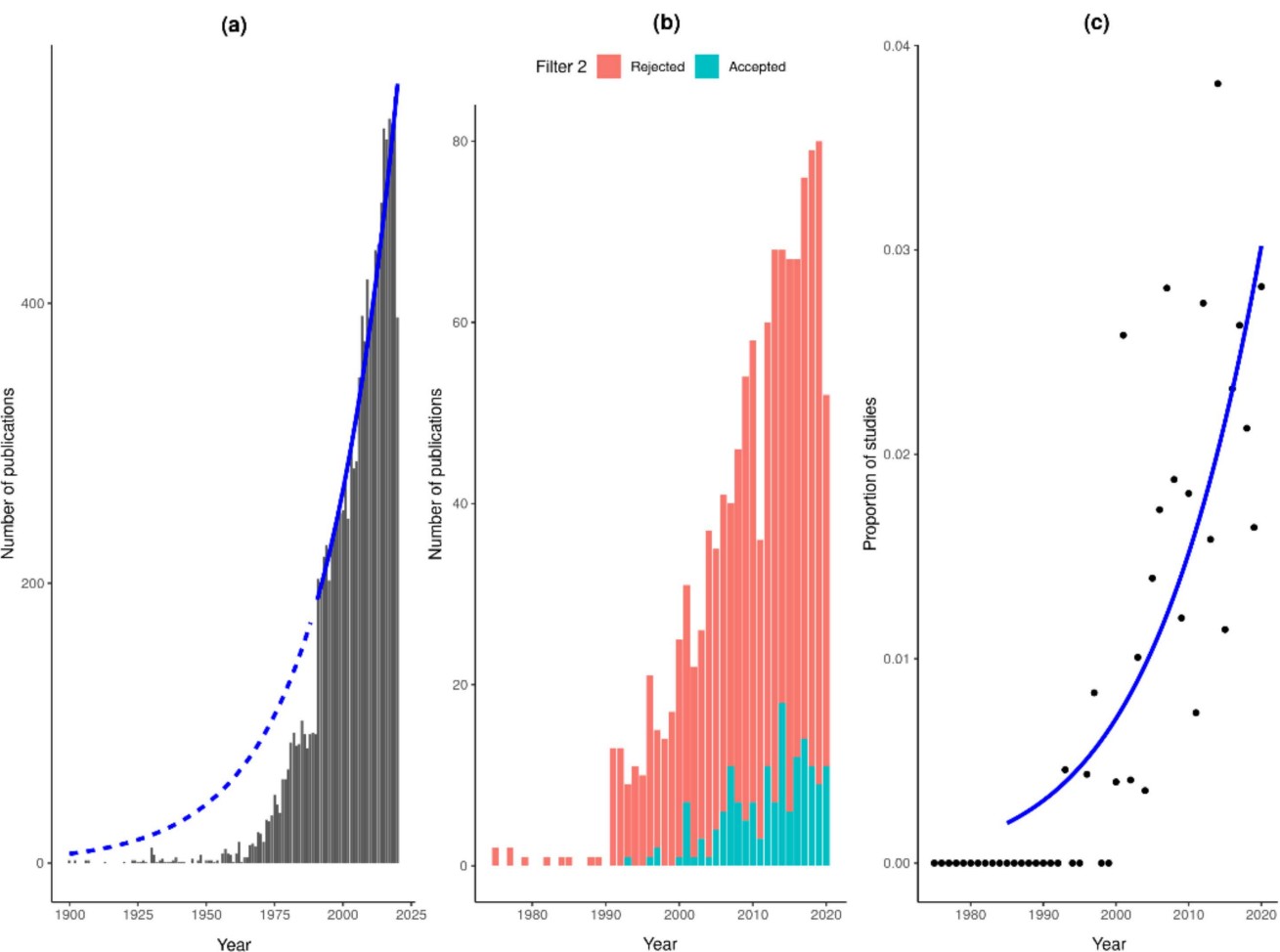

**Figure 1.** Trends in total number and proportion of publications per year. (**a**) Total sample of 12,699 documents with keywords related to parrots from Web of Science; the blue line indicates the modeled exponential increase in total publications per year using a GLM with a Gaussian distribution and log-link fitted to the years 1990–2020 (solid line) and extrapolated to the previous period (dashed line). (**b**) Number of publications filtered by keywords related to species distribution and manually rejected or accepted. (**c**) Proportion of studies on species distributions in relation to the total sample of publications per year (black dots), the blue line represents the modeled increase in proportion using a GLM with binomial distribution and probit link function.

We then applied successive filters to select the most relevant studies to discuss. In the first filter, we performed an automatic screening of the title, abstract, and authors' key words looking for topic specific words: 'distribution', 'change in distribution', 'range assessment', 'niche model*', 'distribution reduction', 'occupancy model*', 'resource selection', or 'species distribution model*'. With this filter we selected 1210 documents (9.5% of the total sample) that likely had information related to parrot distribution. In the second filter, we reviewed title and abstracts, and, if necessary, also the full text, and rejected 939 documents that were evidently off-topic (e.g., different taxonomic groups, not related to species distribution), in addition to opinion articles, and reviews or overviews with no original data or analysis. The remaining 161 documents (1.26% of the total sample) represent the subset of studies that are directly related to distribution of parrot species. The number of documents selected in the first filter rose sharply in 1990, and the first document included in the second filter is from 1993 (Figure 1b). This gap is partly related to the uneven temporal coverage of the WoS database mentioned above, but even focusing on the publications after 1990 we see a significant positive trend in the proportion of publications related to the study of distribution in parrots (Figure 1c). As we discuss below, the onset of this rising trend coincides with first applications of species distribution models to parrot species around the years 2000–2005.

### 2.2. Document Classification

For these 161 documents we made a more detailed assessment of the abstract or full text, evaluating whether they met the following criteria: (a) original analysis (no reviews) of species distribution or related state variables; (b) using statistical modeling approach of any kind; (c) using spatial data (location of records and/or spatial cover of explanatory variables); and (d) making explicit spatial predictions of the state variable (Table S1). Based on these criteria, we found that almost half of the documents (79) were focused on reports of species occurrence records without using any modeling approach (i.e., calculation of extent of occurrence and area of occupancy) or were statistical comparisons of naive occurrence estimates, abundance or resource use between habitat types, sampling areas or discrete regions. For the remaining 82 documents that did apply some methods of species distribution modeling, we summarized information on geographical location, target taxa, topics of research and general application (Table 1).

We aggregated information about the countries where the studies were conducted into five main regions following ISO classification: Africa (Eastern, Northern, Southern and Western Africa), the Americas (North America, Latin America and the Caribbean), Asia, Europe, and Oceania [18]. The list of parrot species reported was normalized using the species list of BirdLife International [19] to unify the species scientific names across documents. We identified whether the focal taxon or taxa were native or non-native parrot species. We classified the main research topic addressed (Behavior, Conservation, Ecology, Evolution, Invasion ecology, and Methodological issues), and split these research topics into general and specific applications (Table 1).

**Table 1.** Major topics addressed in the parrot distribution modelling literature, with general and specific applications and relative examples taken from published case studies. The modelling paradigm and type of data used are shown. ENM = Environmental (or ecological) niche modelling. RFS = Resource Selection Function. SOM = State Observation Models.

| Topics | General Application | Specific Issues | Number of Publications | Paradigm Used | Type of Data | Examples |
|---|---|---|---|---|---|---|
| Behavior | Habitat use related to behavior types | Occurrence of behavior types | 1 | ENM | Literature | [20] |
| | Temporal distribution patterns | Movement related to environmental factors | 1 | ENM | Open access databases | [21] |
| Conservation | Climate change | Change in distribution driven by climate change | 6 | ENM, SOM | Field work; Open access databases; Citizen science project; Literature; Museum collections | [22–24] |
| | | Combined effects of climate and habitat changes | 1 | ENM | Literature; Open access databases | [25] |
| | | Evaluating or forecasting the effect of environmental changes | 1 | ENM | Field work | [26] |
| | Spatial prediction | Effect of conservation actions | 1 | ENM | Field work | [27] |
| | | Identification of priority areas for conservation | 1 | ENM | Literature; Open access databases | [28] |
| | | Resource distribution | 2 | ENM | Field work; Open access databases | [29,30] |
| | Threats monitoring | Change in distribution driven by habitat loss | 7 | ENM, SOM | Field work; Open access databases | [31–36] |
| | | Effect of conservation actions | 1 | RSF | Field work | [37] |
| | | Fragmentation effect | 2 | ENM | Field work | [38,39] |
| | | Input for population models/population viability analysis | 1 | ENM | Field work | [40] |
| | | Threat distribution | 2 | ENM | Field work | [40,41] |
| | | Threat effect on distribution/occupancy | 1 | ENM | Field work | [42] |
| Ecology | Macroecology | Abundance-occupancy relationship | 1 | SOM | Field work | [43] |
| | | Effect of biotic interactions on distribution | 1 | ENM | Open access databases; Museum collections | [11] |
| | | Global distribution patterns of diet type | 1 | ENM | Open access databases | [44] |
| | Relation with environmental variables | Determining areas for survey | 2 | ENM | Field work | [20] |
| | | Identifying breeding habitat | 5 | ENM, RSF | Field work | [45,46] |
| | | Identifying potential habitat | 1 | ENM | Open access databases | [47] |
| | | Inter-annual variability in distribution | 2 | ENM, RSF | Field work | [48] |
| | | Variables affecting distribution | 8 | ENM, SOM, RSF | Field work; Open access databases | [49–53] |
| | Ecological communities | Richness and alpha-diversity | 2 | SOM, RSF | Field work; Open access databases | [54,55] |

**Table 1.** *Cont.*

| Topics | General Application | Specific Issues | Number of Publications | Paradigm Used | Type of Data | Examples |
|---|---|---|---|---|---|---|
| Evolution | Biogeographic patterns | Change in historical distribution | 2 | ENM | Open access databases; Museum collections; Field work; | [56,57] |
| | | Understanding distribution of extincted species | 1 | ENM | Literature; Museum collections | [58] |
| Invasion ecology | Invasion effect | Impacts on native species | 2 | ENM, RSF | Field work; Open access databases | [59,60] |
| | Predictions of invasion risk | Establishment of non-native specie | 3 | ENM | Open access databases | [27,61,62] |
| | | Limitations into invasion risk | 1 | RSF | Field work | [63] |
| | | Niche shift | 3 | ENM | Field work | [21] |
| | | Potential range of invasive species | 5 | ENM | Literature; Field work; Open access databases | [64,65] |
| Methodological issues | Improving estimation | Factors affecting distribution estimation | 1 | SOM | Open access databases; Field work | [66] |
| | Survey methodology biases | Evaluating citizen science data | 1 | ENM | Literature; Open access databases; Citizen science; Field work | [65] |

We evaluated temporal patterns in parrot distribution modeling publications by aggregating the number of published documents by year and by research topic (Figure 2a). Parrots have been recognized as a model group for global macroecology analysis of species distribution [55] and the first application of distribution modelling techniques for any parrot species focused on the ecology of an endangered species [47]. Ecological questions remained the predominant topic between 2005 and 2011, but the diversification of studies led to a balance between more theoretical and applied research questions. Between 2012 to 2015, Ecological studies had a similar cumulative output as Conservation and Invasion ecology combined, but after 2016 Conservation became the most popular topic (36.6% of all studies up to 2020), followed by Ecology (34%) and Invasion ecology (19.5%; Figure 2a).

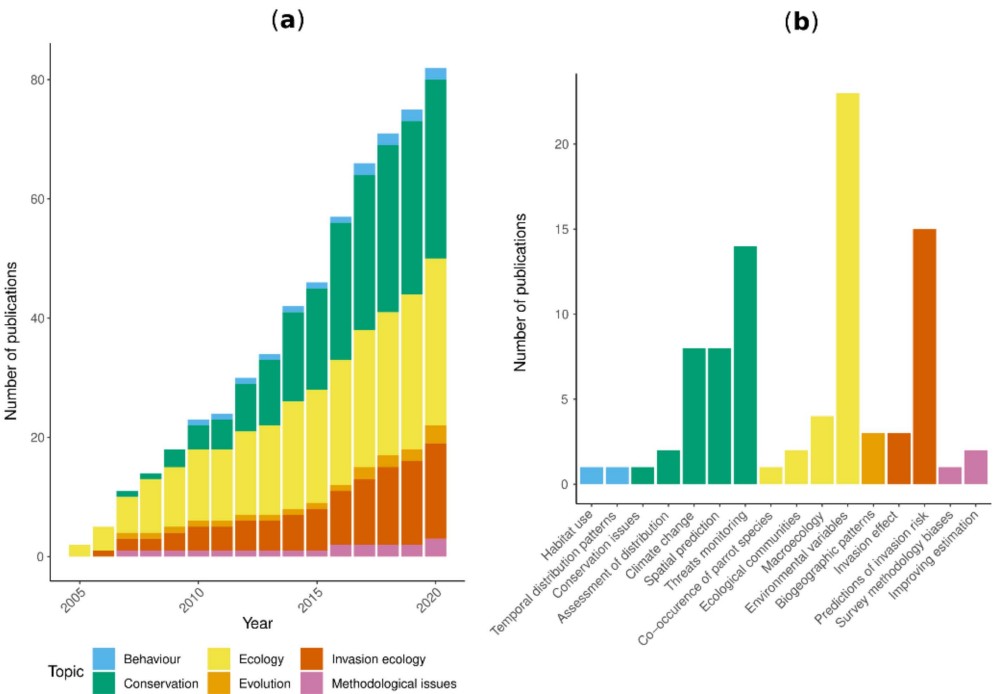

**Figure 2.** Temporal pattern in published literature in parrots' distribution modelling. (**a**) The accumulated number of publications across years is shown for each topic. (**b**) The total number of published documents by general application within each topic.

Within Conservation topics, threat monitoring and climate change were the most frequent applications, whereas Ecological topics were dominated by applications focused on evaluating the relationship between occurrence and environmental variables (Figure 2b, Table 1). Invasion ecology was mainly focused on predicting invasion risk (Figure 2b, Table 1).

To visualize taxonomic patterns, we aggregated the number of documents by genera, research topic and region, and represented these relationships with a bar and bubble plot. The majority of the reviewed literature was focused on species within their native range (86%; Figure 3). The Americas was the region with the highest diversity in applications, but noticeably studies in invasion effects were almost absent. Oceania was the second most diverse region regarding model's applications, but in this region, studies focused on evolution topics were absent. Africa only had a small number of studies in ecology, conservation, and invasion ecology topics. The Americas, Oceania and African regions had studies in both native and non-native species. In contrast, studies from Asia and Europe have been focused exclusively on predicting invasion risk of non-native parrots (Figure 3).

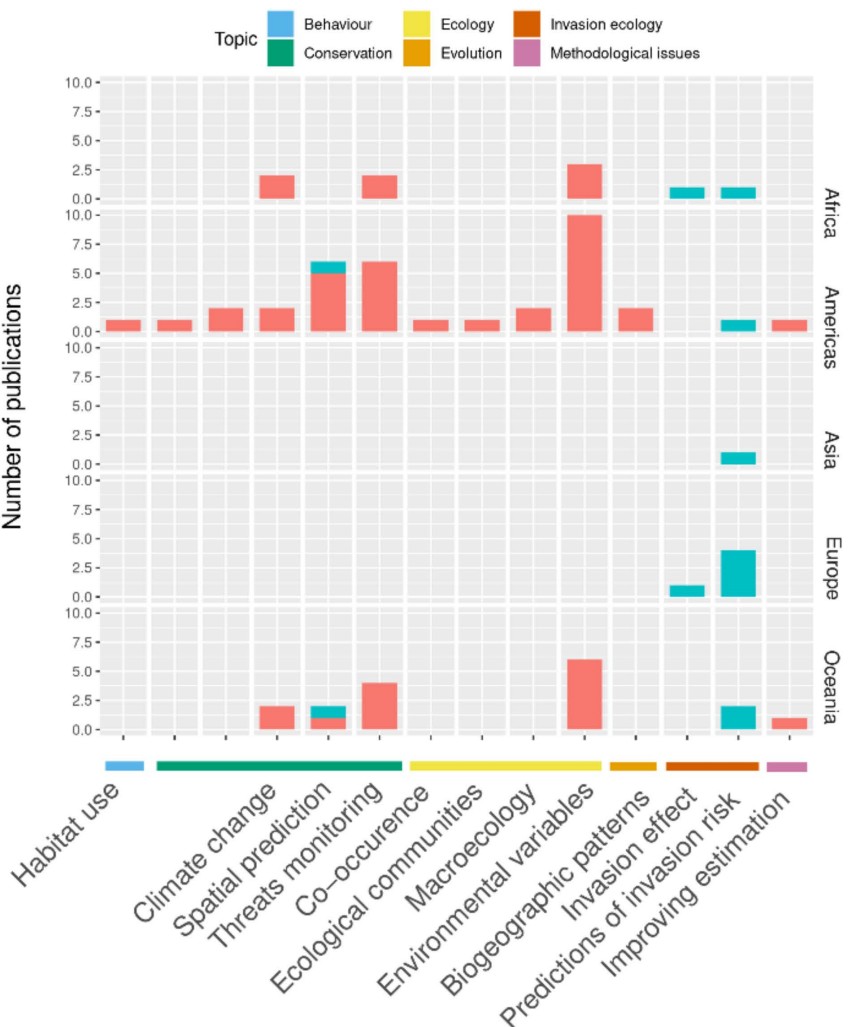

**Figure 3.** Geographical distribution of applications of the parrots' distribution modelling. Number of published documents by region. Documents focused on native species are in red, whereas those focused on non-native species are in blue. Applications are grouped according to the main topics.

We recorded 52 parrot species in the distribution modeling literature. As in other conservation topics [8], parrots' distribution research shows a clear bias toward widespread species such as *Psittacula krameri*, *Myiopsitta monachus*, and *Amazona oratrix.* At the genus level, *Ara* had the highest diversity of applications, whereas *Psittacula* and *Myiopsitta* only have studies focused on invasion ecology (Figure 4). Taxonomic bias can be explained in part by higher availability of occurrence records for species with wide ranges and/or high abundance. However, although *Amazona*, *Psittacula* and *Myiopsitta* genera are among the top ten species with the largest number of occurrence records in GBIF, altogether they only account for 18% of the parrots' occurrence records (9,880,043 records), with other genera such as *Platycercus*, *Cacatua* and *Trichoglossus*, being better represented in the GBIF database [67]. The high impact of invasive species on socio-economic and environmental contexts likely trigger higher interest in describing the establishment and invasion risk of *Psittacula krameri* and *Myiopsitta monachus* (Figure 4). Conversely, the high prevalence of distribution models for *Amazona oratrix* likely results from a combination of the interest raised by their conservation status and a strong and prolific research team in Mexico, where the three main subpopulations of this species occur [39].

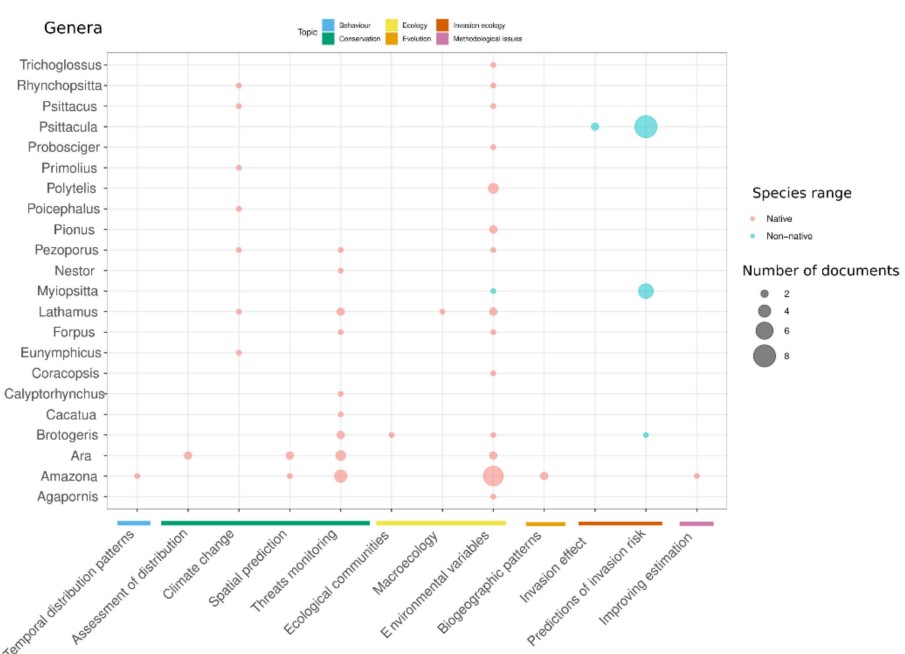

**Figure 4.** Taxonomic patterns of the published literature in parrots' distribution modelling. The number of species by genus and application are shown. The circle size is proportional to the number of documents. Genera are in alphabetical order from bottom to top. Applications are grouped according to the main topics.

In addition to taxonomic bias, we also identified intrinsic geographic biases in the parrots' distribution research, with publications from the America and Oceania regions dominating the research map. This pattern may represent a combination of: (1) a higher diversity of American parrots compared to other regions (233 spp. in the Americas versus 128 spp. in Asia and 129 in Oceania) [1]; and (2) higher scientific capacity in the Americas and Oceania (65% of detected documents; Figure 4). However, our search strategy, which was focused on English and Spanish keywords, likely resulted in an under-representation of literature published in Asian languages. Future efforts should include a wider range of Asian languages to discern whether the observed spatial pattern responds to a detectability problem or to lower publication rate in Asian countries.

## 3. Contributions of Species Distribution Models to Parrots Research

The review of literature showed that the study of parrot species distribution has increased steadily in the past 30 years and has likely been boosted by the widespread use of species distribution models in the past 15 years (Figures 1 and 2). These methods have matured alongside the developments of computer capacity for spatial and statistical analysis, and have become part of a standard toolkits, facilitating their application for a wide range of research questions [68].

In the following sections we discuss the many contributions of species distribution models to the study of ecology, conservation and management of parrot species using illustrative examples identified during our review. All literature reviews are inevitably limited by any biases in the initial selection (search engine, languages and keywords) and the involuntary omissions in the subsequent steps of this process. We have highlighted these biases whenever possible, and stress that we do not attempt to offer an exhaustive account of all subjects.

### 3.1. Distribution Models to Study the Ecology of Parrots

The role of abiotic factors explaining parrot distribution have been to date the most frequent ecological application in parrots' distribution research (Figure 2b, Table 1). This

research conceptually aligns with the environmental (or ecological) niche modelling (ENM) paradigm which focuses on estimating the fundamental niches of species, or ecological requirements of species by relating their known geographic distributions (i.e., occurrence records) to a set of environmental or abiotic variables [12]. Niche models predict habitat suitability or potential distribution, but species may not be using their entire potential habitat due to a range of constraining factors. The two most important natural, non-environmental constraints are biotic relationships and accessibility (Figure 5a) [12]. However, large scale patterns in distribution are often the result of geographic variation in the use of resources at the scale of populations or individual movements (Figure 5b,c). The Resource Selection Function (RSF) paradigm compares the frequency in the use of resources (preference) with the overall distribution of resources in the landscape (availability) [69]. RSF are mostly applied to determine preferences at the scale of individual movements, but for long-lived and highly mobile species this can represent very large geographical areas. Thus, ENM and RSF models may overlap in conceptual and practical terms. For example, an early study of the Superb Parrot (*Polytelis swainsonii*) in Australia focused on describing the bioclimatic envelope of the species as a fixed factor influencing its distribution [70]. In a second study, these authors included plant productivity as a covariate related to the availability of resources and were able to explain seasonal and year-to-year variability in abundance and distribution that was not accounted for by the previous static environmental model [71].

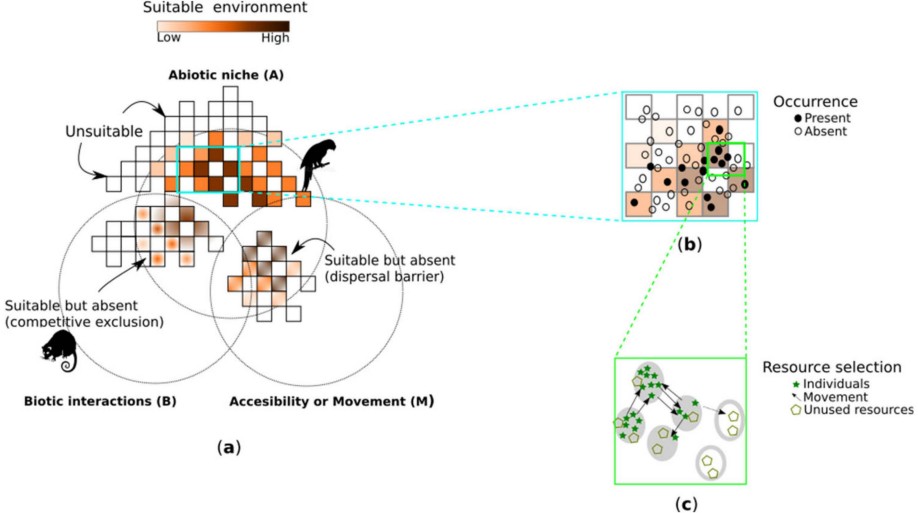

**Figure 5.** The main paradigms for modelling species distribution operate at different hierarchical levels, requiring different types of data and algorithms. (**a**) The niche modelling (ENM) paradigm predicts suitable environmental conditions where species might occur, but although suitable, this habitat cannot be used due to biotic relationships and dispersal barriers constraints. ENM can use presence-only datasets if algorithms such as Maxent, Random Forest, BIOMOD, and GARP are used, but will require "pseudo-absence" data if GLM methods are used instead. (**b**) State-observation models (SOM) work at the population level and predict the occupancy probability conditioned to the probability of detecting the species given it is present. SOM typically requires data from repeated sampling visits (occasions) to a single site during a time frame over which the population is *closed* (e.g., no changes in occupancy between surveys). (**c**) The Resource Selection Function (RSF) paradigm works at individual and species scales and predicts the probability of an animal or species using a certain resource, proportional to the availability of that resource in the environment. RSF requires two types of data: presence records of the focal species (either at individual or species level) and data on the resources available across the study area.

Incorporation of biotic and biogeographic elements in species distribution models allows their application to be extended to a wider range of research questions in biogeography and evolution, community ecology, etc. [13]. Kissling et al. [42] evaluated the roles

of climate and productivity on broad-scale geographical patterns of parrots' richness and whether they show distinct regional differences compared to other frugivore species due to regional patterns of diversification of food plants, niche conservatism and past climate change. Often these broad biogeographical and macroecological patterns can mask more nuanced relationships between species distribution and landscape features at the scale of populations and individuals (Figure 5b). Keighley et al. [49], combined distribution, behavior and genetic information to describe dispersal pathways and barriers for the Palm Cockatoo (*Probosciger aterrimus*) in Australia, and to test hypotheses about key landscape features influencing movement of palm cockatoos throughout their range.

In most cases, local studies rely on field observations of species occurrence to better understand limiting factors and describe temporal changes in occupancy. State-observation models (SOM) use several sampling approaches (multiple visits, multiple observers, distance sampling, etc.) to make joint estimates of a state variable of interest (usually occupancy or abundance) and the observation or detection process (detectability or probability of detection). Normally the SOM paradigm assumes closed populations at each sampling site during the primary sampling periods and explores how species behavior, sampling characteristics and environmental variables might affect detectability across time and geographic space [72]. However, some of these models can be extended to multiple seasons and allow population dynamics between seasons to be studied. Kalle et al. [24] applied dynamic occupancy models to a decade of citizen science-driven presence/absence data on the Cape Parrot (*Poicephalus robustus*) and were able to model recent range dynamics as a function of changing climate conditions and the availability of resources.

### 3.2. The Biotic Component of Parrots' Distribution

Biotic interactions are undoubtedly an important component of species distributions, and these can determine the relationship between parrot species and their habitat. In particular, the relationship between parrots and their food plants works in both ways; the distribution of diet resources contributes to explain the distribution of 11 parrot species in the Cerrado in Brazil [73], and similarly the distribution and density of three large macaws influences the spatial distribution of motacú palm (*Attalea princeps*) in the Bolivian Amazonas savannas. Parrots have a great behavioral plasticity and different species use several strategies to respond to fluctuations in food availability: switch in diet, shift in habitat use and seasonal movements [74]. This plasticity can lead to seasonal (intra-annual) and inter-annual variability in distribution or resource use correlated with changes in spatial indicators. The identification of these patterns requires not only large-scale, but also local scale variables to better describe landscape elements important for parrots' occurrence. For example, key landscape drivers (e.g., woodland structure) determine the occurrence of *Agapornis lilianae* in the mopane woodlands of Zambia [20]; time since fire influences food resources for the Carnaby's Cockatoo (*Calyptorhynchus latirostris*) in fire-prone landscapes in Australia [67]; whereas seasonal use of foraging habitats explains the dynamics of the Swift Parrot (*Lathamus discolor*) in Australia [45,73] and the Maroon-fronted Parrot (*Rhynchopsitta terrisi*) in Mexico [48], in addition to the aforementioned examples of the Cape Parrot and the Superb Parrot in South Africa and Australia [24,70].

Antagonistic relationships can also have a strong influence on species distribution. For example, the study of Engeman et al. [75] suggests that nest site selection in the Puerto Rican Amazon (*Amazona vittata*) is an adaptive response to predation pressure; parrots select nest sites that allow increased avoidance and detection of predators. Moreover, the high occupancy of introduced mammal predators may represent an additional threat to the endangered the Swift Parrot (*Lathamus discolor*) in Australia [41]. Finally, competition and coexistence of parrot species has also been a subject of research in studies using distribution models in areas of high species diversity in the Neotropics [73,76].

### 3.3. Applications of Distribution Models to Conservation Problems

The second most widespread application of species distribution models in parrots is related to threat monitoring for conservation (Figure 2b, Table 1). Monitoring of populations is a basic step of conservation planning and management. Distribution models (specially SOM methods) are used to improve the design of sampling protocols to select sampling areas, optimize probability of detection and reduce uncertainty in estimates of probability of occurrence or other state variables [20,66,77]. For example, some less-conspicuous parrots such as the Blue-fronted Amazon (*Amazona aestiva*) and the Peach-fronted Parakeet (*Eupsittula aurea*) require longer observation times in order to improve detectability [73]. This approach is particularly useful when combining robust sampling designs, automated methods such as camera traps and automatic sound recording, and modeling methods for spatial data analysis [29,42,60].

Distribution models are used extensively to evaluate changes in distribution due to habitat loss [31,32,35,78,79]. Habitat fragmentation is often considered a direct threat to the persistence of species in modified landscapes [38,53]. Plasencia-Vazquez et al. [80] combined spatial prediction with different metrics of landscape fragmentation to explore the relationship between forest fragmentation and the geographic potential distribution of different parrot species in the Yucatan Peninsula, Mexico. The combination of current and historic datasets and appropriate modelling methods for each dataset can be useful to make more explicit tests of changes in distribution. For example, Ferrer-Paris et al. [36] compared historical and contemporary distribution of eight species of Amazon parrots (*Amazona*) in Venezuela and found negative changes in widespread species such as *Amazona amazonica*, and *Amazona ochrocephala*, and rare and patchily-distributed species such as *Amazona barbadensis*.

Climate change was the fourth most frequent application in parrot distribution research (Figure 2b, Table 1). Several studies have focused on predicting changes in distribution driven by climate change [22–24,81]. Assessment of impacts was less studied, but Legault et al. [26] presented a new method for assessing how the population size of the New Caledonian Parakeet (*Cyanoramphus saisseti*), the Horned Parakeet (*Eunymphicus cornutus*), and the Ouvéa Parakeet (*Eunymphicus uvaeensis*) in New Caledonia will change in the future based on the relationship between local abundance and modeled habitat suitability obtained using ecological niche models.

Spatial analysis often reveals unexpected inter-species or species–habitat interactions in modified landscapes that may affect already threatened species. Such is the case of novel predators of the Swift Parrot [40,41], in addition to the relationship between modified fire regimes and habitat use in ground parrots, which can inform management actions [30,42]. Species distribution models are also useful for tracking species introduction and recovery of populations. Recio et al. [37] used GPS tracking to evaluate how a forest-dwelling species, *Nestor meridionalis,* selected habitat within its home ranges, showing that this species moved beyond the predator exclusion fence into urban suburbs. In this example the native forest patches throughout the city facilitated dispersal of individuals between refugee and food sources, and long-term survival will require careful urban planning and management to provide the necessary balance of different elements.

### 3.4. Applications of Distribution Models to Invasion Ecology

Applications related to invasion ecology such as prediction of invasion risk and assessment of invasion effect were also widely detected in the parrot distribution research (Figure 2b, Table 1). Species distributions are dynamic and many species of parrots show a recent natural expansion of their range [24], but in some cases these changes may be confounded by the intervention of humans. Mota-Vargas et al. [57] used ENM methods to compare environmental conditions between historical and recent records of the White-fronted Amazon (*Amazona albifrons*) and discriminate between introductions by humans and natural expansion of its distribution range.

Invasive species and some native species show great adaptability to novel environments, such as urban areas or modified landscapes. Shokuroglou and McCarthy [50] used bird atlas data and Bayesian logistic regression to predict the distribution of the Rainbow Lorikeet (*Trichoglossus moluccanus*) in Melbourne, Australia. Le Louarn et al. [82] compared the use of the urban landscape by a native range-shifting bird (*Corvus monedula*) and an invasive parrot species (*Psittacula krameri*) and found that expansion of the latter is likely driven by its effective ability to exploit urban resources which native species do not exploit. Some tropical islands can become hotspots of exotic species, but not all exotics have the same success as invaders. Falcon and Tremblay [65] analyzed the distribution of parrots in Puerto Rico and found 11 species present only as pets, and at least 29 species in the wild, of which at least 12 were breeding, but most persisted in localized areas and small populations. Only *Brotogeris versicolurus* and *Myiopsitta monachus* showed clear evidence of range expansion.

In most applications the potential risk of invasion or potential distribution of invasive species is predicted from current occurrence records and environmental data layers [64,65,83–85]. Few studies combine these spatial predictions with information about invasion process (i.e., trade, introduction effort, and breeding origin) to improve predictions of environmental suitability and potential niche shifts in the introduced parrots [61,62]. Less prevalent was the evaluation of impacts related to invasion. However, there are some examples of evaluation of economic impacts on agricultural production and human infrastructure [64,83] in addition to some examples of environmental impact through the effects on other animals and competition, and even measurement of the Generic Impact Scoring Scheme [63,83]. Notably, one study goes beyond evaluating the impact of the problem and evaluates the impact of conservation actions such as removing invasive *Trichoglossus moluccanus* [27].

## 4. Challenges and Opportunities for New Research on Parrots' Distribution

The previous examples of applications of species distribution models reflect how research has adapted to address the dynamic and complexity of species distributions. Here we summarize some of the challenges and opportunities for future research.

### 4.1. Social Behavior and Geographic Variability

Linking the distribution of species with social structure of parrot populations can provide better insights into intra-specific variability [86]. Most applications of species distribution models have focused on abiotic covariates or combinations of abiotic and inter-specific interactions (use of resources, predator avoidance, competition), but intra-specific interactions (e.g., dispersal and aggregations of individuals) can influence the connectivity of populations and phenotypical or genotypical variability [87,88].

Many aspects of the ecology of parrots are influenced by their social behavior [86]. Most species exhibit long-lasting pair bonds, occupy large home ranges, and congregate in more or less stable foraging and roosting groups. Active defense of year-long territories is rare, and patterns of ranging and dispersal are often seasonal. Geographical variability of behavioral traits such as vocalizations can serve as an indicator of social structure across a range of scales from individual, pairs and flocks, to populations. For example, landscape resistance models revealed strong effects of isolation by elevation on genetic, repertoire and structural call differentiation in Palm Cockatoos [49]. However, if species are highly aggregated, it is likely that standard species distribution modeling (e.g., Maxent) will not provide adequate predictions. In those cases, regression-based methods that account for spatial structure can be used [88].

### 4.2. From Global to Local

Although many early applications of species distribution models, and particularly ENM, have focused on broad biogeographical or macroecological patterns, current conservation and management applications need more detailed information on changes occurring

at the time scales of one or few generations [89]. RSF implicitly account for movement of individuals and are often linked to seasonality in resource distribution by relying on time series of covariates [90]. Multi-season SOM have explicit means to parametrize changes in state variables (e.g., colonization and extinctions). We expect that future applications will continue to explore the links between static and dynamic components of species distribution, for example, by incorporating sink–source dynamics, connectivity and barriers to explain range contraction, shift and expansion of the distribution in the face of climate and land cover change, or invasion processes in new environments [52].

### 4.3. Automatic Data Collection

Arrays of passive detectors such as camera traps and sound recordings, and the use of unmanned aerial vehicles, have the potential to provide massive streams of data on species presence, abundance and behavior [91–93]. Automatic data collection coupled with machine learning methods to identify species or individuals (image recognition and vocal profiles) have been used to study many emblematic species and to document species diversity [94]. These methods have the great advantage of providing detection histories and allow standardization of sampling protocols and sampling effort across local to regional scales. For example, automated sound recordings were used to model habitat occupancy and post-fire response of the Ground Parrot (*Pezoporus wallicus*) in heathland sites in Australia [42]. An outstanding challenge in this area is the development of virtual platforms for sharing standardized data records that could allow collaboration between research groups and large-scale analysis of spatial and temporal trends [95,96].

### 4.4. Citizen-Science and Socially-Derived Data Collection

Citizen-science has become the main source of species distribution records for many species, especially for birds [97]. User networks such as eBird [98] and iNaturalist [99] provide large platforms for accessing a great volume of data contributed by enthusiastic ornithologists, photographers and other volunteers. Outstanding challenges are the inherent bias in the distribution of observers and the reliability and accuracy of records [100,101].

Less specific social-media platforms can also be used as a source of additional information on species distribution, but they require more active search and filtering of records. Here some records may come from engaged citizen groups and organizations, but many records are accidental or opportunistic. This is, however, a key source for understanding human interaction with wildlife, for example, legal and illegal pet trade, invasion of exotic species, and human–wildlife conflict [102,103]. For example, Mori et al. [61] used several data sources (from eBird to YouTube) to study worldwide patterns of trade and establishment of exotic *Agapornis* parrots.

### 4.5. Cyber Infrastructure for Research

Species distribution models, and especially ENM, have become a mainstream tool for ecological analysis and fully integrated into several analytical workflows. For example, virtual laboratories and cloud applications enable new users to learn these workflows and existing users to explore their potential applications and share their results [104,105]. These platforms offer a seamless integration of data into the analytical workflow. For example, biodiversity records from natural history collections (Global Biodiversity Information Facility, Atlas of Living Australia, etc.) and the main citizen-science networks (eBird, iNaturalist) are imported directly. The focus of these applications is on presence only data, and they are compatible with the ENM paradigm and, to a lesser extent, the RSF paradigm. Similar applications for SOM are missing, probably due to the challenge of dealing with different data structures (detection histories, distance sampling, double observer, etc.) and the more complex statistical and computational context.

*4.6. Supporting Decision Making*

Species distribution models have become an important tool for supporting solutions for on-ground conservation problems. A dialogue between modelers and decision makers increases the opportunities for integration of research outputs into the decision-making process, and will contribute to improve both scientific knowledge and conservation or management outcomes [106].

There are several explicit applications of spatial distribution modeling to support spatial conservation decisions in parrots. Studies in Argentina and Brazil have analyzed the percentage of key parrot habitat covered by protected areas [107–109], and Botero-Delgadillo et al. [65] used models to identify independent conservation units for *Pyrrhura* parakeets in Colombia. The relatively high prevalence of predictive distribution modeling supporting on-ground conservation problems of parrots is particularly encouraging and contrasts with the unclear and less prevalent examples in other conservation contexts [79,106].

## 5. Conclusions

Parrots are a very attractive and interesting taxonomic group for ecological studies, and our review revealed a large variety of studies related to modeling of species distribution. Since the advent of distribution model paradigms, parrots have been used as model taxa to answer macroecological questions. The complexity of parrots' ecology and behavior likely make it an object of research itself, with an increased interest to address more complex questions at different geographical and ecological scales: from environmental niches to biogeographic and biotic interactions [110]. The flexibility of species distribution models and the maturation process throughout the development of algorithms and analytical approaches have allowed their application to applied research questions. Particularly in parrot distribution research, these models have been used to inform conservation action and management for both threatened and invasive species, becoming a remarkable example of how the research–action gap can be reduced and translated into insightful, science-based conservation actions. The integration of species distribution models with other tools in the decision-making process and other data (e.g., landscape metrics, genetic, behavior) could further expand the range of applications and provide a more nuanced understanding of how parrot species are responding to their even more changing landscape and threats.

**Supplementary Materials:** The following are available online at https://www.mdpi.com/article/10.3390/d13120611/s1, Table S1: documents used in the review.

**Author Contributions:** Both authors contributed at the same level in all the stages of manuscript preparation. Conceptualization, J.R.F.-P. and A.S.-M.; methodology, J.R.F.-P. and A.S.-M.; formal analysis, J.R.F.-P. and A.S.-M.; data curation, J.R.F.-P. and A.S.-M.; writing—original draft preparation, J.R.F.-P. and A.S.-M.; writing—review and editing, J.R.F.-P. and A.S.-M. All authors have read and agreed to the published version of the manuscript.

**Funding:** This research received no external funding.

**Conflicts of Interest:** The authors declare no conflict of interest.

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
