# Peer review of "Contributions of Distribution Modelling to the Ecological Study of Psittaciformes"

_diversity, doi:10.3390/d13120611_

Round 1

Reviewer 1 Report

The authors present a review of studies employing distribution modeling (SDM) to the ecological studies of parrots.

I found the topic quite interesting and the manuscript well-written. I commend the authors on the level of detail in the section describing the method and results (including the table and figures) of the literature review.
The section on the contributions of SDM to parrot research provides an excellent overview that should be of value to any researcher considering the application of these methods to study parrots.

Review papers are very useful to compile and condense research that has been done, but they are even more valuable when they point readers to knowledge gaps and new avenues for research in a given topic, and when they guide new researchers (i.e students) on navigating the topic. With that in mind, I would suggest that the authors consider: (I) the inclusion of a section commenting on promising avenues for future applications of SDM research (this is only briefly mentioned in the Conclusions - e.g. integration of SDMs with other tools and data - but I think it merits further elaboration and examples, even if from research on other taxa; (2) another section listing the main difficulties/challenges for the application of SDMs to parrots; and (3) a table to act as a guide on SDM application for parrot research, i.e. summarizing the types of SDMs that are used for within the different research topics, the types of data required, and examples in the literature. Alternatively, the authors could add this information to Table 1. This type of summary is often very useful for students. 

editorial comment: please review section numbering.

Author Response

The authors present a review of studies employing distribution modeling (SDM) to the ecological studies of parrots.

I found the topic quite interesting and the manuscript well-written. I commend the authors on the level of detail in the section describing the method and results (including the table and figures) of the literature review.
The section on the contributions of SDM to parrot research provides an excellent overview that should be of value to any researcher considering the application of these methods to study parrots.

Authors’ answer: Thanks!

Review papers are very useful to compile and condense research that has been done, but they are even more valuable when they point readers to knowledge gaps and new avenues for research in a given topic, and when they guide new researchers (i.e students) on navigating the topic. With that in mind, I would suggest that the authors consider:

(I) the inclusion of a section commenting on promising avenues for future applications of SDM research (this is only briefly mentioned in the Conclusions - e.g. integration of SDMs with other tools and data - but I think it merits further elaboration and examples, even if from research on other taxa; (2) another section listing the main difficulties/challenges for the application of SDMs to parrots; and (3) a table to act as a guide on SDM application for parrot research, i.e. summarizing the types of SDMs that are used for within the different research topics, the types of data required, and examples in the literature. Alternatively, the authors could add this information to Table 1. This type of summary is often very useful for students. 

Authors’ answer: We have included a section 4 in the manuscript to describe challenges and opportunities of SDM in parrots [L445 – L534]. We also have included 2 additional columns in the Table 1 describing paradigm and type of data used to address each specific issue.

editorial comment: please review section numbering.

Authors’ answer: We have corrected the section numbering.

Reviewer 2 Report

The work shows important results on how the science and study of psittacines has been growing over time in different fields. I think it is a relevant work to understand not only what we know, but also what we still have to study. 

I consider that the synthesis made is very relevant.

Author Response

The work shows important results on how the science and study of psittacines has been growing over time in different fields. I think it is a relevant work to understand not only what we know, but also what we still have to study.

I consider that the synthesis made is very relevant.

Authors’ answer: Thanks! We appreciate the review evaluation.

Reviewer 3 Report

  • Did you only collect studies in English? If so, clarify why.
  • Lines 31-32. Please, provide more information, e.g. how many species.
  • The end of the introduction lacks clear aims and predictions. Please specify better why are you conducting this study, and put the paper on a more hypothesis-driven context.
  • Line 82. Only Wos? Why did you avoid Scopus, Google scholar…? You may have missed some works.
  • Methods are ok and well-described
  • Results and Discussion are ok, but I suggest you to add a part on knowledge gaps. For instance, as far as I know, connectivity models to predict population expansions are still missing, whereas they could provide important information and help early detection in new areas of invasion.

Author Response

Lines 31-32. Please, provide more information, e.g. how many species.

Authors’ answer: We have included the number of Psittaciformes species according to the IUCN web site.

The end of the introduction lacks clear aims and predictions. Please specify better why are you conducting this study, and put the paper on a more hypothesis-driven context.

Authors’ answer: We have included several sentences at the final of introduction section address this point.

Did you only collect studies in English? If so, clarify why.

Line 82. Only Wos? Why did you avoid Scopus, Google scholar…? You may have missed some works.

Authors’ answer: We limited our search to one search engine (Web of Science, WoS) and one language (English). At least 88% of the documents were originally published in English, 7.1% in other 21 languages, 5% did not have information on the original language. Thus, all quantitative analysis are limited to a single sample of the literature, but we used them to illustrate general trends and acknowledge their inherent biases and limitations. Although we do not explicitly compare this sample with other sources, we are confident that this search is representative of the overall trends in the scientific literature. Recent reviews have shown that among several academic search engines, WoS is more selective than Dimensions and GoogleScholar and has a high degree of overlap with Scopus. We are aware that the contribution of publications in non-English languages is high and by focusing our search in only English published papers we got a biased sample [L248 - L252]. However, due several non-English journals include abstracts in English we are confident that we were able to get a good representation of topics and applications published on parrot distribution research.

We have added some sentences to clarify this point [L89 – L101; L105 - L107], and references to support these points.

Methods are ok and well-described

Authors’ answer: Thanks !

Results and Discussion are ok, but I suggest you to add a part on knowledge gaps. For instance, as far as I know, connectivity models to predict population expansions are still missing, whereas they could provide important information and help early detection in new areas of invasion.

Authors’ answer: Please, see our reply to reviewer 1. We have included an additional section to address this point.